# Mental Health Interventions among Adolescents in India: A Scoping Review

**DOI:** 10.3390/healthcare10020337

**Published:** 2022-02-10

**Authors:** Devika Mehra, Theophilus Lakiang, Nishtha Kathuria, Manish Kumar, Sunil Mehra, Shantanu Sharma

**Affiliations:** 1MAMTA Health Institute for Mother and Child, New Delhi 110048, India; devika.mehra@mamtahimc.in; 2Public Health Consultant, Medeon Science Park, 205 12 Malmo, Sweden; theo.lakiang@gmail.com (T.L.); nishthak@mamtahimc.in (N.K.); manishkumar@mamtahimc.in (M.K.); dr_mehra@mamtahimc.in (S.M.); 3Department of Clinical Sciences, Lund University, Skane University Hospital, 205 12 Malmo, Sweden

**Keywords:** adolescents, interventions, mental health, programs, school health, scoping review

## Abstract

Early adolescence is the period of the emergence of most mental disorders contributing significantly to the mental health burden globally, including India. The major challenges in India are early identification of mental health problems, treatment gap, lack of professionals, and interventions that address the same. Our review aimed to assess the effectiveness of mental health interventions among adolescents in India. We systematically searched PubMed, PsycINFO, and Cochrane databases and used cross-referencing to review the interventions published from 2010 to 2020. Eleven interventions were included in this review; nine were school-based, one community, and one digital. Most of the school-based programs used a life skills curriculum. Additionally, coping skills and resilience curricula showed improvement in depressive symptoms, cognitive abilities, academic stress, problem-solving, and overall mental well-being. The multi-component whole-school intervention was quite promising and helped in improving the overall school climate and various other mental health outcomes. Hence, school-based programs should be implemented as an entry point for screening mental health problems. However, there is a need for a more comprehensive mental health program in the country for adolescents. Additionally, there is a need to address the gap by conducting more interventions for early and out-of-school adolescents.

## 1. Introduction

Mental health is fundamental to good health and well-being, and it influences social and economic outcomes throughout life [1,2]. Childhood and adolescence are crucial for laying a foundation for healthy development and good mental health [3]. The increasing burden of mental health problems among this population is a growing concern globally [4]. Most mental disorders begin before 25 years of age, more often between 11–18 years [5]. The burden associated with common mental disorders (depressive and anxiety disorders) rises in childhood and peaks in adolescence and early to middle age (10–29 years) [6]. A meta-analysis estimated that the global prevalence for any mental disorder among children and adolescents is 13.4% [7].

Multiple factors are associated with poor mental health among adolescents, including domestic violence, child abuse, bullying, peer pressure, substance abuse, human immunodeficiency virus (HIV) infection, and teenage pregnancy [8]. Transition through stressful and unhealthy adolescence with a high burden of mental disorders can impact their health and well-being later in life [9]. Hence, investment in adolescents delivers a “triple dividend”—improving their health now, enhancing it throughout life, and contributing to the health of future generations [5].

India is home to the largest number of adolescents globally, comprising about a fifth of its population (243 million) [10]. A meta-analysis reports that 6.5% of the community and 23.3% of school children and adolescents have psychiatric disorders [11]. India has the highest youth suicide rate globally, and suicide is the leading cause of mortality in this population [12]. The National Mental Health Survey (2015–2016) reported a 7% prevalence of psychiatric disorders in 13–17 years and was nearly equal among both the genders [13]. The overall impact of mental illnesses on society is enormous, and the awareness about the severity of mental health diseases is very poor. It is critical to note that the economic burden of mental health disorders outweighs other non-communicable diseases. It can be as high as 4% of the gross national product, among which only 2% has been accounted for by treating mentally unhealthy people in developing countries [14]. The overall treatment gap for mental health disorders in India is as high as 90% [13].

India has a National Mental Health Program, which functions at the district level (District Mental Health Program) and is also working towards delivering mental health as a part of integrated primary care within the public healthcare system. The government of India has also started the National Adolescent Health Program (Rashtriya Kishor Swasthya Karyakram) [15], which has mental health as the priority area. Still, there is a huge gap in addressing the mental health needs of adolescents [16], and unfortunately, the situation has worsened since the COVID-19 pandemic. Thus, it makes a crucial case to take mental health at the forefront of the issues that need to be addressed.

With the present burden and effect of mental health disorders among adolescents, it is necessary to identify effective interventions. There is emerging evidence from high-income countries that interventions in different settings and life domains among adolescents contribute to good mental health and prevent mental health problems [8]. Unfortunately, there is a dearth of such mental health interventions in India, and even fewer that have been evaluated [17]. Furthermore, the few systematic reviews that assessed such interventions have analyzed evidence from high-income countries [18,19,20]. To the best of the authors’ knowledge, no such systematic review is available specifically from India. The lack of mental health programs for adolescents in schools and out-of-school adolescents in India may be due to various challenges such as problems in conducting school-based psychosocial programs, adolescent stigma related to mental health problems, and lack of qualified mental health workers, etc. [21]. With the COVID-19 national lockdowns, adolescents have gone through acute and chronic stress because of parental anxiety, disruption of daily routines, increased family violence, and home confinement with little or no access to peers, teachers, or physical activity [22]. Thus, it is an opportune time to explore the paradigm of mental health awareness as a means for combating stigma, enhancing prevention, ensuring early recognition, and stimulating simple and practical interventions within the community [16]. This study aimed to review the effectiveness of various interventions to prevent and manage mental health disorders among adolescents in India.

## 2. Materials and Methods

The main research question of this study was: what are the various community, school, digital, individual, or family-based interventions and their effectiveness in improving mental health and well-being of adolescents in India?

This scoping review was conducted systematically by following Preferred Reporting Items for Systematic Review and Meta-Analysis extension for Scoping Review (PRISMA-ScR) guidelines [23].

### 2.1. Study Selection

Inclusion criteria

Peer-reviewed articles published in the last ten years (January 2010–March 2020) and reporting interventions that targeted adolescent mental health were included. In addition, we included studies that targeted adolescents (aged between 10–19 years). We decided upon a scoping review due to two reasons. First, we had budget and human resource constraints for a complete systematic review. Second, adding >10 years to the search filter may not have given us recent evidence, if we had not found articles in the last ten years on the situation of adolescent health interventions.Types of studies: programs/interventions designed to promote/improve positive mental health among adolescents (The design included pre-post, randomized control trial (RCT), quasi-experimental, mixed-method, and prospective cohort studies) were included in this review.Outcomes: the primary outcomes of interest were mental health and well-being benefits, including indicators of positive mental health such as self-esteem, self-efficacy, coping skills, resilience, emotional well-being; negative mental health such as depression, anxiety, psychological distress, suicidal behaviour; and well-being such as social participation, empowerment, communication, and social support.

The current scoping review focused on four broad categories for mapping the interventions as used in a previous systematic review: school-based interventions, community-based interventions, digital platform-based interventions, and individual-/family-based interventions [8].

Exclusion criteria

Reports, conference papers, commentaries, editorials, meta-analyses, systematic reviews, clinical studies, and studies that are still in the formative phase were excluded from this study.Studies that were not written in EnglishStudies that were not conducted on humans.

### 2.2. Search Strategy

A web-based search was conducted through PubMed, PsycINFO, and Cochrane databases, and cross-references were used to identify the studies using various combinations of search terms. The following search terms formed the search strategy, which were used in various combinations: (mental health OR depression OR suicide OR anxiety OR “substance abuse” OR “drug use” OR “addiction” OR “behavioural problems” OR “eating disorder” OR “anorexia” OR “bulimia” OR “PTSD” OR “OCD”) AND (“community health” OR “mass media” OR ehealth OR “school health” OR “digital health” or “mHealth” OR family) AND (project OR promotion OR implementation OR trial OR evaluation OR intervention OR intervention study OR program*) AND (adolescents OR “adolescent*” OR young people OR youth). Table 1 shows the search terms used for the database search. Cross-references (screening reference list of included studies) were used to add other relevant studies to the review. The search was filtered for studies conducted in India and conducted on humans.

### 2.3. Screening of Studies

All articles were screened against a set of pre-determined criteria mentioned above (Section 2.1). Titles and abstracts were screened independently by two reviewers, and studies were taken forward only when both reviewers reached a mutual agreement for inclusion. For full-text screening, only those studies approved by both authors were included in the review. A final matrix was then developed in consensus with all three researchers involved in the literature review.

### 2.4. Data Extraction and Synthesis

Data were extracted using a standardized, pre-tested data extraction form by one reviewer and verified by two experts. A thematic narrative synthesis of included articles was done to summarize the findings of each intervention. Similar interventions were grouped, followed by a narrative synthesis of each reported intervention regarding the interventions’ feasibility, components of interventions, reasons behind particular interventions that worked and did not work, and comparisons between these different interventions.

### 2.5. Data Abstraction and Quality Assessment

The appraisal was done for all the papers after the full-text screening. The quality of evidence of each article included in the review was assessed independently by two reviewers using the Joanna Briggs Institute Quality Appraisal tool for this review’s research study designs [22]. The detailed Joanna Briggs Institute Critical Appraisal Checklist for all designs, Quasi-experimental study [24] (Appendix A), RCT [24] (Appendix A), Qualitative study [25] (Appendix A) and cohort studies [24] (Appendix A) are mentioned in the study. There was no disagreement between the two reviewers. A total of 11 papers were finally included in this review.

## 3. Results

### 3.1. Characteristics of the Interventions

The flow chart of the articles included and excluded is shown below in Figure 1.

An overview of the key characteristics of the interventions, including study design, sample characteristics, contents of the intervention, main findings, and outcomes, are summarized in Table 2. A detailed description of the interventions (including the type of intervention, length of the intervention and follow-up period, details of the intervention characteristics, the background of the trainers, and tools used for evaluation) is provided in Appendix A.

### 3.2. Narrative Synthesis of the Interventions Included in the Review

#### 3.2.1. Objectives, Design, Sample and Setting Characteristics

A majority of the school health interventions in the review focused on different life-skills education programs for improving adolescents’ well-being, including coping skills programs, self-esteem, academic stress management, self-awareness, adjustment, motivation, internal loci of control, and self-determination, interpersonal relations, and communication skills to address various mental health issues [26,27,28,32,34]. The school interventions also used health education modules, resilience curriculums, and multi-component whole school programs to improve the mental well-being of adolescents [29,30,31,33]. We found only one digital intervention that aimed at the acceptability and feasibility of an mhealth intervention, which was basically text messages for promoting a helpline for young women to promote positive mental health [36]. Additionally, only one community-based intervention was found that addressed the acceptability and feasibility of a population-based intervention to promote youth’s health, conducted in communities through the peer leaders in the rural areas and by the trained teachers in the schools (urban areas) [35].

Out of the eleven studies included in this review, eight used quantitative study designs [26,27,28,29,30,31,32,34], which were a mix of pre-post evaluations, cluster-RCT or RCT, prospective cohort, two studies adopted a mixed-method analysis [33,35], and only one study used a qualitative study design [36]. The majority of the samples in this review were conducted on school-going adolescents in the age group of 13–20 years from 6th–12th grade; only one community-based intervention included 16–24-year-olds. The sample size of the intervention groups varied from 13 to 5316 participants. The intervention programs were conducted in both government and private schools in rural and urban areas. Study participants from government schools were primarily from low to middle socioeconomic status, while those studying in private schools were from middle to high socioeconomic status. The settings where the interventions were conducted were also quite varied: Northern (Delhi), Southern (Bangalore, Kerala, and Chennai), Western (Goa), and Eastern (Kolkata and Bihar) regions of the country.

#### 3.2.2. Intervention Components and Protocol

Most of the life skills programs in school-based interventions included developmental issues in adolescent components followed by mental health issues. In addition to psychological skills such as social learning, positive behavior change, and empowerment, themes such as hygiene, substance abuse, sexuality education, and nutrition were covered in three interventions in this review [27,29,31]. The concept of beginning with basic life skills education was the preferred choice. Moreover, teaching adolescents about life skills is believed to improve their adaptive behaviour to deal effectively with the demands, challenges, and various stressors of everyday life.

With regard to the background of trainers, seven studies were conducted by researchers, interventions were delivered by trained workforces, which included teachers called SEHER (Strengthening Evidence Based on School based Interventions) Mitra (a friend who is a lay counsellor), school counsellors, female therapists, trained peer-leaders, community medicine professionals who were trained by different mental health professionals such as psychologists, psychiatrists, social workers, psychotherapists, and life skills educators [28,29,30,31,33,34,35]. In three studies, interventions were delivered by the researchers themselves [26,27,32], whereas, only in one study, the intervention information was delivered in the form of digital messages [36]. However, in all the studies, the intervention content for the delivery during the intervention session was developed or examined by professionals with a psychosocial background who were mental health professionals or human behavioural professionals.

The program protocol of the interventions conducted in schools ensured that the day-to-day functioning of classes was not to be disturbed. At the same time, systems were put in place (e.g., compulsory attendance) to see that students attended the programs. The duration of the interventions varied across different programs, with some being conducted on two successive days in each week to one month in a year, with each of the sessions varying between 45 to 120 min. The format of the sessions conducted was either individually, in pairs, or in small groups of 10–15 students. Other interventions that did not mention the format utilized in the intervention, it may be presumed, included the entire class as a whole. To sustain the interests and motivation of adolescents for these programs, several interactive and participatory strategies were designed, for example, games, competitions, co-curricular activities, etc. Most interventions utilized available infrastructure of the school, teachers, and parents to implement programs and ensure their continuation. Only one digital mhealth intervention used text messages as a medium to deliver positive mental health and health tips to them [36]. The community-based intervention also followed the protocol [35].

#### 3.2.3. Intensity and Extent of the Interventions with Its Effect on the Outcomes

In this review, there were variations in the extent and intensity of the interventions, which ranged from 28 h to 2 years. There were only two studies in our review which were conducted for a very short period; one was a digital intervention which was conducted for a month with a text-based mobile messaging service for a helpline, which showed an effect in terms of acceptability and feasibility (user satisfaction for the support services) of the mhealth service [36]. The other study was conducted in Kolkata among adolescents for two months that showed an effect only at the knowledge and attitudinal level with respect to certain psychosomatic aspects [29]. Hence, studies for a shorter duration did not generate much evidence regarding behavioural change of the participants.

There were two studies in this review that were conducted in schools of Bangalore. Both had an eight-week coping skills program in schools, which showed clinically significant results with a reduction in the severity of depressive symptoms, negative cognition, academic stress, and social problem-solving except coping skills for the study that was conducted in 2014 [26,32]. Another study in our review that focused on life skills education programs showed improvement in coping skills, pupil-teacher adjustment, and psychosocial behaviour [28]. There was no difference between the groups after adjustment with parents and peers and on psychopathology assessed by the Strengths and Difficulties Questionnaire. A life skills intervention in schools for improving self-esteem and psychological well-being of adolescents with psychosocial problems showed a statistically significant correlation between self-esteem and its different dimensions (personal, social, and general) within seven sessions (28 h) of life skills education [34].

A two-year life skills intervention significantly improved resilience in the intervention group compared to the control group [27]. There was also significant improvement (marginal mean increment) in internal locus of control, self-determination, and reduced pathological behaviours in adolescents three months post the intervention. An RCT study conducted for a 5-month duration showed a significant improvement in emotional resilience, self-efficacy, social-emotional assets, psychological well-being, and social well-being. However,, the effect of depression was not captured in the study [30]. Therefore, rigorous interventions are required to show visible improvements in major psychological illnesses such as depression.

A recent RCT SEHER assessed a multi-component, whole-school intervention delivered by lay counselors (seher mitra) and school teachers for one and a half years, which showed improvement in the school climate, depression, bullying, attitude towards gender equity, violence victimization, and violence perpetration [31]. No significant effect was seen when the intervention was delivered by school teachers, but a positive shift was seen in the overall well-being of the adolescents when delivered by lay counselors for a longer duration.

The PRIDE (Premium for Adolescents) project is a trans-diagnostic study based on the stepped care system in Indian schools, which showed that six week-long sessions scheduled with psychologists were not feasible, as participants struggled to use the resource materials outside the guidance sessions. A modified Pilot 2 was implemented by less experienced counselors for 3–4 weeks, where the outcomes were maintained with enhanced feasibility and acceptability, and the impact score of the intervention was also observed to be higher than the previous study [33]. Hence, the frequency and quantity of the interventions in school-based settings for upscaling should go through a stringent review process to address the common mental health problems among adolescents.

The current review encountered only one community-based intervention with a follow-up of 18 months for promoting the overall health of youth [35]. Community-based peer education and health information materials were given in rural settings, and teacher training was conducted in urban settings. In both the intervention settings, probable depression was lower and led to higher knowledge and a positive shift in the attitudes towards sexual reproductive health. The results from the rural settings showed that there were fewer menstrual complaints, higher levels of help-seeking for sexual and reproductive health complaints by women, and an increase in knowledge and attitudes about emotional health and substance use. The urban sample reported significantly lower levels of substance use, suicidal behaviour, sexual abuse, and sexual and reproductive health complaints. Hence, multi-component interventions for a longer duration in the community and institutional settings showed positive results for the overall health promotion of adolescents.

## 4. Discussion

This review sought to assess the effectiveness of the mental health interventions designed for adolescents in India. We found only 11 interventions in three categories, namely school-based (n = 9), community (n = 1), and digital (n = 1). We did not find any intervention on an individual/family level in the review. Our review highlighted the dearth of interventions on mental health among adolescents, and very few have been rigorously evaluated.

### 4.1. School Interventions

Findings from the school-based interventions indicate that more than half of the interventions were life and coping skills programs-based for achieving positive mental health outcomes. Other interventions were carried out at school that included peer education programs, teacher training programs, health education curricula, and resilience curricula. School mental health programs generally have been implemented on a limited and fragmented scale, despite adolescent mental health being a priority area for development in the National Adolescent Health Program (Rashtriya Kishor Swasthya Karyakram [15]. The interventions largely on life and coping skills were delivered in various modalities, which showed a significant positive effect on students’ emotional and behavioural well-being, including depression, anxiety, resilience, and coping skills. The programs that have been reviewed and presented in this article conclusively indicate that a multi-pronged intervention strategy results in improved adolescent mental health outcomes. Evidence shows that life skills education integrated into the school mental health program using existing resources is a cost-effective way to deliver the interventions [28]. This corroborating evidence of life skills programs conducted in other low- to middle-income countries (LMIC) such as Cambodia has shown improvement in the overall mental health of adolescents, particularly among boys with high-risk behaviours [37].

Global evidence from a systematic review conducted in LMICs and high-income countries highlights the importance of teacher training and the provision of ongoing support during program implementation [3]. To date, most of the studies from LMICs have focused on promoting adolescent mental health and improving the school environment. However, there is evidence that shows its scope should be extended to include therapeutic interventions tailored to the resources available and in line with the key preferences of stakeholders, most critically including adolescents themselves [38]. These programs can serve as a naturalistic base from which individualized programs can be developed for adolescents with intensive needs, avoiding the stigmatization that often arises when individualized programs are implemented in isolation from other program goals [39].

There is robust evidence from substantial reviews conducted in high-income countries showing a positive effect on multi-component interventions (i.e., adopting social competence approach and developing a supportive environment) when compared to interventions that focus on specific health behaviours [40,41,42]. Therefore, integration of multi-component programs within the whole school approach [43] based on generic social and emotional skills training, addressing common risk and protective factors delivered within a supportive school environment along with parents and community, has the potential to reach out to larger populations with fewer resources [3]. There are very few interventions from India that have started to consider a whole school approach. One such intervention was found in our review that is a rigorously evaluated multi-component whole school intervention delivered by lay counsellors (SEHER project) in school settings, which showed significant improvement in the overall school environment, depression, attitude towards gender equality, and violence [31]. Most of the studies that adopted the whole school approach for mental health programs are from high-income countries [44].

Another interesting intervention in our study was the PRIDE project, a ‘blueprint’ based on global evidence contextualized in an Indian setting [33]. It is a trans-diagnostic, low-intensity, psychological intervention with problem-solving as the primary element. It took two years to design this intervention with various iterations to test its feasibility and acceptability in India’s government-run schools. The pilot was run in schools for 3–4 weeks by less experienced counsellors, which showed positive results. The authors suggest that the intervention has the potential to be a cost-effective first-line trans-diagnostic study for addressing common mental health disorders in India. The RCT trial aims to provide a definitive test of the effectiveness of the intervention and the specific impact of sensitization activities on referral generation. There are inter-linked research efforts that aim to shape the final specifications and implementation of a comprehensive stepped-care program intended to reduce the adolescent mental health burden at scale in India.

### 4.2. Community Interventions

School and community-based interventions are more relevant to LMICs where there is a shortage of mental health professionals. But, in this review, we found only one community-based intervention conducted among adolescents, with one arm implemented in a rural community setting delivered through peer leaders, and the other delivered in an urban setting through trained teachers [35]. The review showed that peer education was a feasible approach (in rural settings). In contrast, in educational institutions, it showed acceptability but not feasibility. Peer leaders were unable to deliver the intervention through structured programs, but delivered it through discussions or acting as role models for behavioural change. We did not find many mental health interventions in adolescents, but community-based intervention with mental health promotion does have an effect. There is evidence from multi-component community-based studies conducted in South Africa, such as the stepping stones on young people, which was actually an HIV program with a mental health component that showed a reduction in depression levels of women and intimate partner violence [45]. Another study was a collaboration of HIV adolescent mental health programs in South Africa, the Collaborative HIV Adolescent Mental Health Program (CHAMPSA) conducted on adolescents, which showed a significant increase in caregivers’ communication skills, monitoring of children, and primary social networks [46].

Therefore, there is a need for more community-based interventions, essentially to reach out-of-school adolescents. One of the critical reasons for the lack of community-based mental health interventions is primarily the dearth of mental health professionals in community settings, including primary health centres and district hospitals. There is growing evidence for focusing on training the ‘lay health workers’ to conduct mental health interventions in the communities, but most such interventions in India are conducted for adults. An example of such an intervention is Atmiyata, which is based on creating community champions in rural India (Maharashtra) to address mental health problems [47]. Vigorous interventions for out-of-school adolescents are necessary to address their mental health challenges and focus on their well-being.

### 4.3. Digital Interventions

The review found only one digital intervention, which was a pilot to assess the acceptability and feasibility of the mobile text message service to promote a helpline for addressing young women’s mental health problems in urban slums. It showed user satisfaction and provided emotional support for mental health promotion and prevention. India is still emerging with its technological advances in tele-mental health, mobile apps, training mental health professionals, and information delivery to the general public [36]. These innovations need to be evaluated for gender sensitivity and confidentiality in providing authentic and reliable support services so as to create more evidence for upscaling digital initiatives. One such initiative in this review is the PRIDE project [33], which is developing POD (identifying “Problems,” generating “Options,” and creating a “Do it” plan) adventures, a blended problem-solving game-based intervention for adolescents with or at risk for anxiety, depression, and conduct difficulties in India [48].

On the other hand, there are digital mental health interventions from high-income countries that have used cognitive behavior therapy and have shown a positive effect on adolescents’ and emerging adults’ anxiety and depression symptoms [49,50]. We need to draw lessons from such studies regarding how effectively we can integrate digital platforms to reach out to adolescents to achieve better mental health outcomes.

The heterogeneity of the interventions from this review in terms of programs, content, delivery, duration, and sample size makes it difficult to draw conclusions regarding the effectiveness of the interventions. However, there is evidence that interventions that were given for a longer duration have a positive effect on the mental health and well-being of adolescents. A systematic review on mental health interventions showed that interventions with a short duration, less frequent, and fewer sessions did not show a very significant effect on improving the mental well-being of the participants compared to interventions that were more structured and for a longer duration [3]. The majority of the interventions were delivered to adolescents aged 13 years and older. Thus, there is a dearth of interventions that focus on early adolescents aged between 10–12 years. It is evident from previous research that early intervention is recognized as an effective mental health promotion and prevention strategy.

### 4.4. Strength and Limitations

The systematic review has some limitations which may have impacted the study. Due to the time and resource limitations, we investigated only scientific databases and have not included grey literature studies. We searched for studies that were published in the last 10 years only, which may have limited the scope of this study. Because there is a dearth of mental health interventions in adolescents currently, there would be very few interventions from more than 10 years ago, so we focused on recent data. Limitations related to the selection criteria of the studies included in the review may have impacted the validity of the study. Studies that were not related to common mental health disorders among adolescents were not included in the study. Studies that did not have any mental health intervention, mental health outcome, prevalence-based studies, studies in the formative phase, and RCT protocols were not included in this review.

We provided a narrative synthesis and conducted a quality assessment of the studies included in this review, but we did not generate meta-analysis data. Despite these limitations, the studies included in this review demonstrate that high quality and effective mental health promotion interventions and their evaluation through well-designed research studies are feasible in LMIC settings. The study results may be affected by publication bias for positive findings as the negative findings may not have been reported. Some of the studies that were included in the review did not provide validation of the outcome measures. Additionally, we hardly found any studies that were from the smaller cities of India.

## 5. Conclusions

The findings from this review show that mental health promotion interventions for adolescents were effective in school-based settings. We found promising evidence from interventions delivered using a multi-component whole school approach and life skills curriculum to improve mental health outcomes. We identified the feasibility and effectiveness of integrating mental health with other issues, such as sexual and reproductive health, nutrition, HIV, and substance use in school and community settings, and improving the knowledge of adolescents on these issues along with improving their overall mental well-being.

The findings from a very promising SEHER trial show that intervention, when delivered by teachers, was not effective, and when delivered by trained lay counsellors, had an effect on the overall school climate for depression and anxiety. Using the existing school settings and the school staff, involving stakeholders and community health workers (lay counsellors) for delivering the interventions was found to be cost-effective. However, there is a need for intensive interventions in early adolescents, which were not considered in the majority of the interventions.

We observed a paucity of community and digital interventions addressing adolescents. Digital interventions, including mhealth and telehealth, can act as efficient means of delivering mental health information and services to adolescents, more so in COVID times. These platforms can be a way forward for those in need or can provide primary prevention to those at the risk of developing mental health problems. The review noticed limited interventions that were systematically scaled up to serve the needs of adolescents.

There is a need for further research on mental health issues through longitudinal study designs targeting multi-level systems, so that there is better evidence for scale-ups. There should also be a focus on strategies for the reduction of stigma, which hinders individuals from accessing mental health services, despite the need. It is important that adolescents can utilize services through mechanisms apart from community sensitization and educational trainings and more innovative approaches such as digital platforms etc., which can build their capacity for self-reliance and resilience to individually deal with mental health problems. Additionally, there is a need for the use of contextualized and validated tools in local settings.

## Figures and Tables

**Figure 1 healthcare-10-00337-f001:**
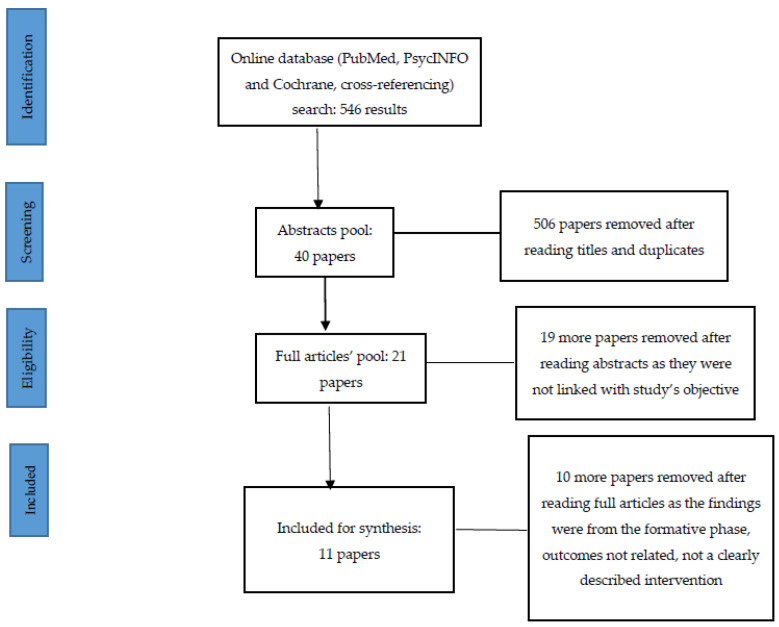
PRISMA Flow Chart.

**Table 1 healthcare-10-00337-t001:** The search terms used for the database search.

Mental Health Terms	Population	Setting	Intervention Terms
Mental Health	Adolescents	Digital health	Project
Substance use	Young people	School health	Promotion
Drug use	Youth	Community health	Implementation
Addiction		Family	Trial
Behavioural problems		Mass media	Evaluation
Eating disorder		ehealth	Intervention
Anorexia			Intervention study
Bulimia			Program
PTSD *			
OCD **			

* PTSD: Post Traumatic Stress Disorder; ** OCD: Obsessive compulsive disorder.

**Table 2 healthcare-10-00337-t002:** Overview of characteristics of mental health interventions in India.

Author	Sample Characteristics	Study Design	Contents of the Intervention	Main Findings	Outcomes
School-based Interventions					
Singhal, et al. 2014 [26]	Sample: I: 13; C: 6 Age: 13–18 years (Grade 9th) Site- Bangalore (3 urbans schools)	Intervention study with pre- and post-design	I: 8-weekly program in schools -Coping Skills Program for adolescents at-risk of depression ^§^ -Change from negative thinking to positive thinking -Academic stress management -Skills to deal with conflicts in interpersonal relationships -Depression Prevention Course C: Psycho-educatory interactive session	I: Children’s Depression Inventory score was significantly higher in the baseline means depressive cognition was higher than the control group (*p* < 0.05) Pre-post comparison: Reduction in the frequency and severity of depressive symptoms and negative cognition Pre to Follow-up: Significant reduction in depression score, depressive cognition, academic stress and improvement in social problem solving C: No effect/change	-Significant reduction in depressive symptoms, negative cognitions, and academic stress -Increase in problem solving and coping skills -Coping skills program useful to reduce or delay the incidence of depression in at-risk adolescents
Sarkar, et al. 2017 [27]	Sample: I: 381; C: 361 Age- 11–17 years (Grade 6–9th) Site- Purulia, West Bengal (2 schools with majority of participants from rural and tribal areas)	Quasi-experimental using solomon four group design ^β^	I: Weekly sessions (on 2-successive days; sessions ranging from 45–120 min) -Basic life skills ^⁑^ -Specific health interventions that include: motivation, discipline, nutrition, health and hygiene, relationship, self-awareness, sexuality understanding body and mind and social responsibility ^¶^ C: No intervention	Significant positive effect on resilience adjusted Odds ratio (aOR) = 11.2 (95% CI = 10.6–11.9), 14.6% higher resilience, improved internal health locus of control, self-determination, and reduced pathological behaviour in the intervention groups	-Improved resilience -Improved self-determination and reduced pathological behavior -Reduced gap in resilience between tribal and non-tribal adolescents
Srikala, et al. 2010 [28]	Sample: I: 605; C: 423 Age: 14–16 years (Grade 8–10th) Site: Bangalore rural district (Chennapatna) and Udupi district (2 schools from each site))	Pre-post evaluation (Intervention and control sites)	I: Life skills education (Min: 5; Max: 16; Average: 10 sessions) included ^α^: -Critical and creative thinking -Decision making and problem-solving -Communication skills and interpersonal relations -Coping with emotions and stress -Self-awareness and empathy C: Regular civic/moral/value education classes (1–2 sessions/week)	Significant difference in self-esteem (*p* = 0.002), perceived adequate coping (*p* = 0.000), better adjustment generally, especially with teachers (*p* = 0.000), in school (*p* = 0.001) and pro-social behaviour between the intervention and the control group in the post-test evaluation. Students in the program showed better outcome in all the aspects in comparison to the students not in program. Teachers who were trained as life skill educators also perceived positive changes in the classroom behaviour and interaction of students.	-Students in the program better adjusted to the school and teachers -Student perceived that they are capable of coping with issues with better self-esteem
Das, et al. 2010 [29]	Sample size-282 female students from 3 schools Age-13–19 years Site- North Kolkata	Pre-post intervention Study	I: Knowledge of adolescent health through education sessions spread over 2 weeks and each lasting 2 h -Causes of health problems -Physical changes during adolescent boys and girls -Psychological problems-Substance abuse -Sex education and sex differences -Sessions were followed by clinical examination by doctors from CM and OBG specialists	A pre-post intervention comparison shows an improvement in the ‘knowledge’ and a ‘positive attitude’ on adolescent health concerning certain psychosomatic aspects. The first post-test was administered at the end of each session followed by the second post-test after 4 weeks.	-Improved adolescent health knowledge -Significant improvement in attitude towards sex education
Leventhal, et al. 2015 [30]	Sample size- I: 1752 at the start and 1681 at 5th month C: 756 at the start and 706 at 5th month Age- 13–14 years Site: 3 rural blocks of Patna district, Bihar	Randomised Control Trial (RCT): 4 arms ^¥^ Arm_1: RC Arm_2: HC Arm_3: RC + HC Control: SC	† RC: Intervention sessions include listening skills, character strengths, life stories and goals, identifying emotions, worry, stress and fear, group problem-solving, forgiveness and apologies, working together to design and carry out projects to increase peace in their own or others’ lives. Program facilitators facilitated weekly sessions in pairs with groups of approximately 12–15 girls over 5 months (1 h per week for 23 weeks) during school hours using manuals, curricula, sessions, etc. and delivered in Hindi, the local language.	1. Psychosocial assets: RC had a positive effect on emotional resilience, self-efficacy, and social-emotional assets, with a significantly high score, observed in the intervention group (*p*’s < 0.01; ES’s = 0.46, 0.58, 0.45, respectively). In the control group, only one outcome (emotional resilience) improved significantly, and self-efficacy. and social-emotional assets decreased significantly over time. 2. Psychosocial well-being: RC had a positive effect on positive psychological well-being and social well-being with increased scores observed in the intervention group (*p*’s < 0.01; ES’s = 0.18, 0.17, respectively). Anxiety scores also increased in the intervention group (*p* = 0.025; ES = 0.15). Girls in the control condition significantly improved in depression and anxiety, and significantly decreased in social well-being.	-Improved emotional resilience, self-efficacy, psychological and social well-being in the intervention group
Shinde, et al. 2020 [31]	*Sample size*- Arm_1: 25 schools (*n* = 5316) Arm_2: 24 schools (*n* = 4475) Arm_3: 25 schools *(n* = 4623) *Age*- 13–15 years *Site*- Nalanda district, Bihar	Cluster randomised control trial: 3 arms Arm_1: Govt-run AEP Arm_2: Govt-run AEP + SEHER by SM Arm_3: Govt-run AEP + SEHER by TSM	AEP **: Delivered by school teachers; classroom-based sessions, 16 h sessionsSEHER ***: -Promoting social skills -Knowledge on health and risk behaviours -Problem-solving skills -Positive and responsible relationships -Gender and sexuality -Prevention of HIV & STDs. -Substance abuse -Engaging school community (adolescents, teachers, and parents) in school-level decision-making process -Access to factual knowledge about health and risk behaviours to the school community -Enhances problem-solving skills	Participants in the *SM* Group had higher school climate scores at endpoint (mean beyond blue school climate questionnaire) [adjusted mean difference = 7.33; 95% CI: 6.60–8.06; *p* < 0.001) compared with those in the control group, and on most secondary outcomes. In the group of *TSM* v/s Control, there was no effect found. The effect of SM-delivered intervention was larger for most secondary outcomes (depression, attitudes towards gender equality, bullying etc.) after 17 months follow up compared to 8 months: school climate (effect size [ES; 95% CI] = 2.23 [1.97–2.50] versus 1.88 [1.44–2.32], *p* < 0.001) and on the secondary outcome, suggesting incremental benefits with an extended intervention.	-Improvements in school climate, depression, bullying, attitude towards gender equity, violence victimization, and violence perpetration
Singhal, et al. 2018 [32]	Sample size: I: 65 C: 55 Age: 13–18 years Site: Bangalore (across 2 urban schools)	Two-group comparison design with assessments at baseline, post-intervention, and 3-months; no-contact follow-up	I: 8-weekly school-based coping skills program for adolescents with sub-clinical depression -Program in same-gender groups of 4–8 adolescents each on depressive symptoms, negative cognitions, academic stress, social problem solving, and coping skills C: 1 interactive psycho-educatory session	75–80% of the adolescents in the intervention group achieved recovery on all measures ^⁑⁑^ and the recovery was more in the intervention than control group (statistically significant difference) 13–63% evidenced improvement and 3–22% achieved a functional status in the intervention group. None of the adolescents showed clinically significant deterioration in either group. The large effect size was reported on all measures at post-intervention and follow-up assessment.	-Significant reductions in depressive symptoms, negative cognitions, and academic stress. -Increase in problem solving and coping skills
Michelson, et al. 2019 [33]	Sample size: Pilot 1: *n* = 45 Pilot 2: *n*= 39 Age: 13–20 years Site: New Delhi schools	Prospective cohort design	¶¶ Pilot 1: Problem-solving steps (‘SONGS’) + re-designed printed self-help materials + workbook + handoutsProblem-solving delivered through guided self-help: help included female psychologists, counselling assistants + expert-led supervision and peer group supervision; classroom sessions of problem-solving and whole-school sensitization. Duration: 6 weeks Pilot 2: Problem-solving steps (‘POD’) and 3 psychoeducational ‘POD-booklets’ on problem-solving + emotion-focused coping strategies and full-colour POD. Problem-solving delivered through active, counsellor-led face-to-face intervention; help included counsellors + peer group supervision meetings + weekly telephone calls with supervisors; re-designed classroom sessions with emphasis on self-identification and normalization of mental health problems; whole-school sensitization. Duration: rapid delivery over 3–4 weeks	Mean service satisfaction scores ranged from good to excellent (Mean = 28.55; SD = 2.48; range = 22–32) Pilot 1 Acceptability: 84.4% completed the intervention Referral rate: 6.8%Feasibility: Average number of sessions completed = 3.82 (SD = 0.73; range = 3–5) ^γ^ Impact: Improved clinical outcomes and moderate to large effects found on Strengths and Difficulty Questionnaire (SDQ) Total Diffniculties score (effect size* = 0.79; 95% CI = 0.42–1.15), Impact score (effect size* = 1.99; 95% CI = 1.43–2.54), Youth top problems (YTP) score (effect size* = 1.89; 95% CI = 1.35–2.42) Pilot 2 *Acceptability*: 74.4% completed the intervention *Feasibility*: 69.0% received the maximum dose (Mean = 4.90 sessions; SD = 0.31; range: 4–5) Referral rate 17.5% *Impact:* Improved clinical outcomes with moderate to large effects found on SDQ Total Difficulties score (effect size* = 1.29 (95% CI = 0.79–1.78), Impact score (effect size * = 1.17 (95% CI = 0.69–1.64) YTP (effect size * = 0.91 (95% CI = 0.47–1.33)	- Adolescents were able to do solve problems effectively.
Azeez A, 2015 [34]	Sample size- 30 (boys: 22; girls: 8) Age: 15–19 years Site: Rural Palakkad district, Kerala	A single group pre-and post-test quasi-experimental design	-Life skills education (7 sessions covering 10 core life skills and emphasis on psychological well-being and self-esteem) -A total of 28 h intervention with ice-breaks, role plays, games, group discussions, and relaxation techniques	The psychological well-being of the participants significantly improved after the intervention (*p* < 0.001) and the overall self-esteem also showed a significant association (*p* < 0.001) after the intervention. The different dimensions ^§§^ of self-esteem also showed an effect apart from life-item. There was no difference between both the groups as it is a single pre-test and post-test group.	-Enhanced mental health and well-being of adolescents
Community-based Interventions					
Balaji, et al. 2011 [35]	I: One rural and urban community C: One rural and urban community matched on urbanization and socio-economic development Sample size study: Baseline: R: 1803; U: 1860 Follow up: R: 1620; U: 1942 Age: 16–24 years Site: Goa	Exploratory controlled evaluation of the intervention in two pairs of urban and rural communities and semi-structured interviews	I: Educational institution-based peer education, teacher training, community peer education program and use of health information materials (Delivered by social workers, psychologists, and peer educators) for 12-months C: Received the intervention after the study	There was a statistically significant difference in adverse outcomes at follow up between the intervention and comparison group in both rural and urban communities. Rural communities: Reduction in probable depression (OR 0.33, CI 0.23–0.48) and physical violence (OR 0.29, CI 0.15–0.57), and increase in the knowledge and attitude about emotional health (OR 1.57, CI 1.18–2.10) and substance abuse (OR 3.83, CI 2.77–5.31) in the intervention group compared to the comparison group Urban Communities: Reduction in probable depression (OR 0.57, CI 0.41–0.79), physical violence (OR 0.59, CI 0.40–0.87), sexual abuse experience (OR 0.19, CI 0.09–0.41), substance abuse (OR 0.63, CI 0.45–0.89), and suicidal behaviour (OR 0.38, CI 0.17–0.84) in the intervention group compared to the comparison group Increase in the knowledge and attitudes about RSH (reproductive sexual health) by 25.1% in the intervention arm, whereas in the comparison arm, this decreased by nearly 6% (OR = 1.46, 95% CI = 1.09–1.97).	-Probable depression and perpetration of physical violence decreased -Enhanced knowledge and attitudes about reproductive sexual health - Reduction in suicidal behaviour and substance use
Digital Interventions					
Chandra, et al. 2014 [36]	*Sample size*: 40 girls *Age:* 16–18 years *Site:* Urban slums of Bangalore	Qualitative assessment (Focused group discussions)	-Text messages on positive mental health tips or helpline information -Helpline message asked girls to message or call back if they felt like talking to someone when emotionally upset	62.5% called back to ask about the mental health services and felt good about the services; 57.5% messaged back about their feelings. 62% felt supported by the messages.	-Psychological general well-being was enhanced

Abbreviations: aOR: Adjusted Odds Ratio; AEP: Adolescent Education Program; C: Control group; CI: Confidence Interval; CM: Community Medicine; ES’s: standardized Effect Size; OBG: Obstetrics and Gynecology; OR: Odds Ratio; I: Intervention group; R: Rural; RSH: Reproductive and Sexual health; SEHER: Strengthening Evidence Based on School-based Intervention) for promoting adolescent health program; SM: SEHER Mitra or friend who was a lay counsellor; TSM: SEHER Mitra who was a teacher; U: Urban; ^§^ Children at-risk of depression have elevated but sub-clinical symptoms of depression defined by cut-off scores on the Children’s Depression Inventory, and the Center for Epidemiological Studies-Depression Scale for Children; ^⁑^ The basic life skills content was adapted from the intervention module of Adolescent Girl Empowerment Program developed by the Population Council; ^¶^ Specific health interventions content was adapted from the Life Skill Education module developed by NIMHANS, Bangalore, India; ^β^ Solomon four group design is a research method that involves four groups (Intervention with pre-and post-test, and only post-test; similarly, no intervention with pre-and post-test and only post-test); ^α^ Life Skill Education program delivered through participatory learning methods of games, debates, role-plays, and group discussions; ^¥^ RC: Resilience Curriculum; HC: an adolescent physical health curriculum; SC: school-as-usual control; ^†^ Girls First Resilience Curriculum (RC) is a low-cost, flexible, and scalable curriculum for middle-school girls in low- and middle-income countries. It is meant for girls in marginalized, high-poverty settings that aims to strengthen emotional resilience (including coping skills, adaptability, and persistence), self-efficacy; and social-emotional assets (including social skills and beliefs about helping others in the community). This will help girls improve psychological well-being (greater life satisfaction and positive affect; lower levels of anxiety and depression) and social well-being (stronger connections with peers). ** AEP was delivered by trained teachers through classroom-based sessions on the process of growing up, establishing positive and responsible relationships, gender and sexuality, and prevention of HIV, other sexually transmitted infections, and substance use to grade 9 and 11 students during the first year of the study. *** SEHER: The SEHER multicomponent, whole-school intervention emphasised the importance of positive school climate that aims to strengthen supportive relationships among school community members, a sense of belonging to the school, a participative school environment, and student commitment at the academic levels. The intervention focused on three levels: whole-school, group, and individual levels; ^⁑⁑^ Measures employed in the study included Children’s Depression Inventory, Centre for Epidemiological Studies-Depression Scale for Children, Children’s Automatic Thoughts Scale, Scale for Assessing Academic Stress, Social Problem-solving Inventory, Adolescent Coping Orientation to Problems Experienced Inventory; ^§§^ Self-esteem was assessed through Culture freeSelf Esteem Inventory, which has 40 items with a Yes or No response. The scale has four dimensions, namely social self-esteem, general self-esteem, life items and personal self-esteem; * Effect size measured as Cohen’s d. ^γ^ Impact was assessed through outcome tools, such as The Strengths and Difficulties Questionnaire (SDQ), which is a 25-item self-report measure of youth mental health, A Total Difficulties score is derived and an Impact Supplement measures associated distress and functional impairment, with an additional descriptive item on the chronicity of difficulties, The Youth Top Problems (YTP), which is an idiographic measure that identifies, prioritises and scores adolescents’ three main problem. ^¶¶^ Problem-solving steps ‘SONGS’ included to identify a problem situation(S), identify options(O) to solve the problem, narrow down the options by considering pros and cons(N), go for it by trying out the best option(G), sit back and evaluate the outcome (S); Problem-solving steps ‘POD’ included: identifying and prioritising distressing/impairing problems (‘Problem identification’), generating and selecting coping options to modify the identified problem directly (problem-focused strategies), and/or to modifying the associated stress response (emotion-focused strategies) (‘Option generation’), implementing and evaluating the outcome of this strategy (‘Do it’).

## Data Availability

Not relevant to the present study.

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
