# Peer review of "Mental Health Interventions among Adolescents in India: A Scoping Review"

_healthcare, 2022, doi:10.3390/healthcare10020337_

Round 1
Reviewer 1 Report
The manuscript is very well organised and sufficiently well-founded. It is a scientific literature review based on the PRISMA protocol, which was adequately followed and described in the work.
Although the authors have analysed a small number of scientific articles in the review (n=11), they have done this work very cautiously and clearly. They were able to extract relevant information, offering results that contribute to the knowledge of mental health care strategies for adolescents in India.
I venture to say that the authors could be less detailed in writing the qualitative summaries of the findings of their literature review. On the other hand, as we progressed in reading the manuscript, we were able to appropriate details highlighted by the authors.
I consider that the manuscript can be published as presented. There is a writing style (marked by prolixity), but it does not compromise the content presented. It is suggested an effort of greater analytical synthesis in future manuscripts by the authors, to avoid repetition of technical-scientific arguments in the work.
Author Response
Comments: I venture to say that the authors could be less detailed in writing the qualitative summaries of the findings of their literature review. On the other hand, as we progressed in reading the manuscript, we were able to appropriate details highlighted by the authors.
Reply: Thanks, reviewer, for highlighting this. We have removed some lines from the results section which seemed extra and could be skipped. We deleted lines 209 and 210 and 239-247.
Reviewer 2 Report
This paper presents the results of the review aimed to assess the effectiveness of mental health interventions among adolescents in India.
The topic of the manuscript is within the scope of the Journal. I find that manuscript is relevant to the aims of Healthcare journal.
TITLE
I think that the title of the article is accurate.
ABSTRACT
Abstract reflects the work done and the conclusions drawn.
INTRODUCTION
The introduction creates a niche for the current investigation by showing a hole in the literature and discussing how it intends to fill it. Some clarifications are however needed.
I think, all scientific categories must be correctly defined, especially adolescence. Most definitions of the more recently introduced term adolescence range from about 10 to 24 years of age but not from 15 to 24 years.
The aim of the study needs to be revised. It is not correct to talk about youth instead of adolescents.
METHOD
Some clarifications are needed.
I suppose that age range of adolescence must be correctly defined and explained.
The sentence “The age range of 15-24 years captures adolescence” must be corrected because authors analyse articles where age of adolescence is from 13 years.
RESULTS
The technique of data analyses seems appropriate.
DISCUSSION
I suppose that not only the limitations of the study could be defined but also further possible research directions, basing on the results of this research, could be explained.
TO SUM UP I think the author(s) need to make the recommended corrections.
Author Response
Comment 1: I think, all scientific categories must be correctly defined, especially adolescence. Most definitions of the more recently introduced term adolescence range from about 10 to 24 years of age but not from 15 to 24 years.
Reply: The definition of adolescents has been revised in the manuscript as 10-19 years in the Line 95-96. Page 2.
Comment 2: The aim of the study needs to be revised. It is not correct to talk about youth instead of adolescents.
Reply: The aim of the study has been revised with only adolescents, and youth have been removed from the aim now in Line 81 Page 2.
METHOD
Some clarifications are needed.
Comment 3: I suppose that age range of adolescence must be correctly defined and explained. The sentence “The age range of 15-24 years captures adolescence” must be corrected because authors analyse articles where age of adolescence is from 13 years.
Reply: We are sorry this was a gross mistake from our end, The age range of adolescents has now been corrected to 10-19 years in the manuscript.
DISCUSSION
Comment 4. I suppose that not only the limitations of the study could be defined but also further possible research directions, basing on the results of this research, could be explained.
Reply: Point well taken we have looked into the limitations section again and have added 1 more limitation there and have included a paragraph on possible further research in Line 459-462.
Reviewer 3 Report
This scoping review aimed to review the effectiveness of various interventions to prevent and manage mental health disorders among adolescents and youth in India. This is a timely and generally well-written paper. There are a number of questions (minors) that arise in the reading of this paper that are reported below to improve this manuscript.
1/I suggested to removed figure 1. PRISMA Flow Chart in the “result” section
2/ Lines 166 to 179 were redundant with Figure 1. This paragraph could be summarized or deleted.
3/ Lines 161 to 164 should introduce “Materials and Methods” section
Author Response
Reviewer 3
This scoping review aimed to review the effectiveness of various interventions to prevent and manage mental health disorders among adolescents and youth in India. This is a timely and generally well-written paper. There are a number of questions (minors) that arise in the reading of this paper that are reported below to improve this manuscript.
Comment 1: I suggested to removed figure 1. PRISMA Flow Chart in the “result” section
Reply: The Prisma flow chart has now been added to the results section
Comment 2: Lines 166 to 179 were redundant with Figure 1. This paragraph could be summarized or deleted.
Reply: This paragraph has now been deleted from the paper
Comment 3: Lines 161 to 164 should introduce “Materials and Methods” section
Reply: These lines which were the research question of the study has now been shifted to the material and method section as suggested.